# Influence of springtime atmospheric circulation types on the distribution of air pollutants in the Arctic

Manu Anna Thomas[1], Abhay Devasthale[1], Tiina Nygård[2]

[1]Research and Development, Swedish Meteorological and Hydrological Institute (SMHI), Folkborgsvägen 17, Norrköping, 60176, Sweden.
[2]Polar Meteorology and Climatology Group, Finnish Meteorological Institute (FMI), Finland.

*Correspondence to*: Manu Anna Thomas (manu.thomas@smhi.se)

**Abstract.** The transport and distribution of short-lived climate forcers in the Arctic is influenced by the prevailing atmospheric circulation patterns. Understanding the coupling between pollutant distribution and dominant atmospheric circulation types is therefore important, not least to understand the processes governing the local processing of pollutants in the Arctic, but also to test the fidelity of chemistry transport models to simulate the transport from the southerly latitudes. Here, we use a combination of satellite based and reanalysis datasets spanning over 12 years (2007-2018) and investigate the concentrations of $NO_2$, $O_3$, CO and aerosols and their co-variability during 8 different atmospheric circulation types in the spring season (March, April and May) over the Arctic. We carried out a Self-Organizing Maps analysis of mean sea level pressure to derive these circulation types. Although almost all pollutants investigated here show statistically significant sensitivity to the circulation types, $NO_2$ exhibits the strongest sensitivity among them. The circulation types with low-pressure systems located over the northeast Atlantic show a clear enhancement of $NO_2$ and aerosol optical depths (AOD) in the European Arctic. The $O_3$ concentrations are, however, decreased. The free tropospheric CO is increased over the Arctic during such events. The circulation types with atmospheric blocking over Greenland and northern Scandinavia show the opposite signal in which the $NO_2$ concentrations are decreased and AODs are smaller than the climatological values. The $O_3$ concentrations are, however, increased and the free tropospheric CO decreased during such events.

The study provides the most comprehensive assessment so far of the sensitivity of springtime pollutant distribution to the atmospheric circulation types in the Arctic and also provides an observational basis for the evaluation of chemistry transport models.

## 1 Introduction

The transport of anthropogenic pollutants from the southerly latitudes has many implications for the Arctic (Law and Stohl, 2007; Quinn et al., 2008; Shindell et al., 2008; Arnold et al., 2016; Willis et al., 2018; Abbatt et al., 2019; Schmale et al., 2021). At daily to weekly scales, the pollutants could exert an impact on the direct radiative forcing, thereby conditioning the

atmospheric thermodynamics and influencing the surface energy budget. The transport of short-lived climate forcers (SLCFs), in particular, absorbing aerosols such as black carbon, is important in this context. The SLCFs can modulate the energy budget at shorter time scales, thereby possibly influencing the seasonal sea-ice evolution. Apart from their direct radiative effects, the SLCFs and other anthropogenic pollutants can also influence the cloud properties, exerting the so-called indirect effects. At climate time-scales, while mitigating the effects of increased carbon dioxide ($CO_2$) and methane ($CH_4$) could take many decades to even few hundred years, the regulation of SLCFs is considered as one of the effective strategies that could be implemented meanwhile to curb the overall impact of increasing greenhouse gases.

The Arctic Ocean is a very special region in this context, not only due to its geography and unique nature of environmental conditions, but also, due to the absence of any major sources of anthropogenic pollution in the central Arctic. The pollution sources are located either in the coastal zones or in the mid-latitude regions. This means that the net effect of SLCFs and the efficacy of their reduction measures depends heavily on the atmospheric transport and the prevailing local atmospheric circulation patterns, which could either dampen or favour the intended effects. This is also an area of research, where there exists a large knowledge gap currently. The uncertainties in model simulations of the impact of SLCFs on the Arctic are therefore high, limiting the design and assessment of the relevant reduction policies.

Pollutant transport to the Arctic occurs nearly all year round, and this transport is heavily influenced by large scale atmospheric circulation and various dynamical mechanisms, for example, cyclones, location of the storm track, high‑latitude blockings, North Atlantic and Arctic Oscillations etc. (Duncan and Bey, 2004; Messori et al. 2018; Papritz and Dunn-Sigouin,2020), as well as the local environmental and meteorological conditions (for example, structure of the atmospheric boundary layer, temperature and humidity inversions, the state of the sea-ice, clouds etc.) during different times of the year. In spring, the meteorological conditions in the Arctic are also usually more diverse than in the winter or the summer months, and the photochemistry begins to play an important role as the solar illumination conditions improve. The polar dome (Bozem et al. 2019), isolating cold air masses in the lower troposphere in the high Arctic from the rest of the Arctic, starts to weaken in spring, allowing for more frequent exchange of air masses between the high Arctic and the lower latitudes. In addition to other anthropogenic sources, the pollutants from biomass burning are also being carried to the Arctic in spring (Stohl et al., 2007; Warneke et al., 2009, 2010). A host of studies have rightfully pointed out the existence, implications and importance of Arctic haze in shaping the Arctic weather and climate in the springtime. Hence, the spring season is a good test bed to investigate the coupling of prevailing weather states and the pollutant distribution in the Arctic. Furthermore, purely from the observational perspective, the availability of satellite-based observations from the sensors that rely on the solar channels increases in spring, as the improved solar illumination conditions allow the retrievals of trace gases.

In light of the reasons mentioned above, it is understandable that a number of major campaigns have been carried out in spring, providing valuable data and characterizing pollutant variability in relation to the transport and local meteorological

conditions. The aircraft measurements, ARCTAS (Arctic Research of the Composition of the Troposphere from Aircraft and Satellites) and ARCPAC (Aerosol, Radiation, and Cloud Processes affecting Arctic Climate), among others, that were carried out as part of the POLARCAT (Polar Study using Aircraft, Remote Sensing, Surface Measurements and Models, of Climate, Chemistry, Aerosols and Transport) campaign for the spring and summer of 2008, provided a wealth of knowledge on Arctic pollution, the transport pathways and climate impacts (Law et al., 2014). This campaign period coincided with a variety of meteorological conditions that affected the transport of different pollutants into the Arctic. For example, ARCTAS data constrained with AIRS CO observations revealed that Arctic pollutants were dominated by European anthropogenic sources from surface to the free troposphere in some cases and by Asian anthropogenic sources above 2 km (Fisher et al., 2010, Jacob et al., 2010). The Asian transport pathways are mainly via the warm conveyor belts (Stohl, 2006). Low altitude ARCPAC flights also revealed increased pollutant concentrations, such as BC, throughout the Arctic atmospheric column during early spring of 2008, indicating accumulation of pollutants during the winter months due to lower temperatures, lack of solar radiation and stable stratification (Spackman et al., 2010). Also, Warneke et al., (2009) identified a significant influx of pollutants into Alaska from the forest fires in Russia and the agricultural burning in Asia. Modelling studies that followed these measurements estimated a reduction (0.8% in spring) in snow albedo over the Arctic owing to BC deposition originating from Russian fires (Wang et al., 2011).

The large-scale descent and stratospheric intrusions also play a role in the observed enhancement of pollutants. For example, BrO concentrations at lower levels were also noted to be enhanced as a result of intrusions of lower stratospheric air into the troposphere (Jacob et al., 2010). The enhanced BrO is also closely linked to frontal lifting in a polar cyclone in spring (Blechschmidt et al., 2016). Despite a negative ENSO year, Arctic weather was strongly influenced by the Eurasian/North American anthropogenic or boreal fires (Brock et al., 2011; McNaughton et al., 2011) resulting in increased concentrations of CO and aerosol loading (van der Werf et al. 2010; de Villiers et al. 2010; Schmale et al. 2011; Quennehen et al. 2011; Di Pierro et al. 2013). Based on the aircraft measurements, Wespes et al., (2012) inferred that up to respectively 45 % and 60 % of the total $O_3$ and $HNO_3$ observed below 400 hPa over the Arctic were of European origin which is transported via northward and westerly trans-Siberian pathways. The contribution of these pollutants from the Asian and North American sectors to the Arctic was much weaker. Most recently, Thomas et al (2019) investigated the dependency of aerosol vertical distribution on the degree of atmospheric stability in the Arctic during winter and spring using the satellite observations. They argued that the observed dependency can be explained by the dominance of pollution transport within the boundary layer during winter and in the free troposphere during spring.

It is evident from the previous studies that a detailed assessment of the co-variability of atmospheric circulation types and pollutants is needed in the Arctic; a) to fully grasp the coupling between local meteorology, pollutant distribution and long-range transport in the Arctic, and b) to improve the representation of such co-variability and coupling in the models.

Such assessment will also help to evaluate and better constraint the existing chemistry transport models as well as fully coupled Earth System models. In the present study, we therefore pose and seek answers to the following scientific questions:

1. Which typical atmospheric circulation types (CTs) prevail in the Arctic during springtime and what are the typical meteorological conditions associated with them?

2. How do these circulation types influence the distribution of trace gases such as $NO_2$, $O_3$ and CO?

3. Is there a distinguishable signal in the aerosol distribution during these circulation types?

## 2 Observational datasets and methodology

The satellite-based datasets of $NO_2$, CO and aerosols for March, April and May months from 2007 to 2018 are analyzed in this study. These are respectively based on retrievals from the Ozone Monitoring Instrument (OMI) onboard the NASA's Aura satellite, the hyperspectral Atmospheric Infrared Sounder (AIRS) instrument onboard the Aqua satellite and the Cloud and Aerosol Lidar with Orthogonal Polarization instrument onboard the CALIPSO satellite. All three satellites belong to NASA's Afternoon Train (A-Train) convoy of satellites, thus providing simultaneous observations in space and time. The ozone dataset is obtained from the Copernicus Atmospheric Monitoring Service (CAMS) reanalysis.

We analysed AIRS Standard Daily IR-Only Version 7 product for the 500 hPa CO retrievals and OMI OMNO2d Version 3 product for the total column $NO_2$ retrievals. The surface conditions and cloud cover play an important role in data sampling. These issues are taken into account before applying a criteria for the selection of the data for each of these species and AOD. In this study, the all-sky OMI NO2 retrievals are used and the quality control is applied similar to the previous studies (e.g. Thomas and Devasthale, 2017). A sensitivity study, wherein we investigated the $NO_2$ anomalies during all-sky and clear-sky conditions (not shown here) is carried out. Though there are some differences in the magnitude of the anomalies, the overall NO2 response and patterns remain robust. In the case of CO, the hyperspectral capability of AIRS allows relatively accurate retrievals even under the presence of partial cloudiness. Therefore, in this study, we have considered cloud cover up to 70%. The high latitude regions are often characterized by the presence of either low level boundary layer clouds or the high thin cirrus clouds, both of which do not significantly affect the AIRS retrievals in the free troposphere at 500 hPa. It is also worth pointing out that previous studies have shown that the circulation patterns that favor pollution transport into the Arctic are also associated with the transport of heat and moisture into the Arctic, which in turn leads to increased cloudiness (Devasthale et al., 2020; Thomas et al., 2019; Johansson et al., 2017). Hence, to capture these most realistic scenarios, stringent thresholds on cloud cover are relaxed in the analysis. By imposing a strict threshold on cloud cover (for example, analysing only clear-sky conditions to ensure the best quality retrievals) would introduce unrealistic clear-sky biases. To investigate the tropospheric aerosol optical depths (AOD), the CALIPSO Level 2, standard aerosol profile product version 4.2 available at 5 km horizontal resolution is used (CAL_LID_L2_05kmAPro-Standard-V4-20). In the case of CALIPSO

APro product, we select data only when the Cloud-Aerosol-Discrimination Score is between and equal to (-50, -100) and when the Extinction Quality flag is 0, 1 or 2. For all satellite products, the data from the ascending passes (daytime conditions) are used. We analyse the retrievals designated TqJ (joint temperature and humidity retrievals) in the AIRS product as they are of best quality and are suitable for process and climate studies. These datasets have been previously used to study the meteorological conditions and pollution variability in the high latitude regions, including the Arctic (Devasthale et al., 2011; Devasthale and Thomas, 2012; Thomas and Devasthale, 2014; Devasthale et al., 2016; Thomas and Devasthale, 2017; Thomas et al., 2019).

Furthermore, we analysed $O_3$ at 925 hPa from CAMS since the focus is on the near-surface $O_3$ and also, since the satellite-based observations of the lower tropospheric ozone are either not reliable or available. The validation of the ozone CAMS reanalysis product is carried out extensively using ground based measurements (TOAR database for surface ozone (Schulz et al., 2017a; 2017b) and ozonesondes globally (Inness., et al., 2019; Huijnen, et al., 2020). CAMS assimilation system makes use of data from SCIAMACHY, MIPAS, OMI, MLS, GOME-2, and SBUV/2 for $O_3$. Even though the surface ozone is primarily model based, upgrades in the CAMS chemical data assimilation system, assimilated measurements etc has improved the near surface estimates.

The dominant CTs in the Arctic in spring are identified and clustered by applying the Self-Organizing Map (SOM) method, developed by Kohonen (2001). The SOM method uses unsupervised learning to determine generalized patterns in input data, and the method has previously been utilized to statistically cluster synoptic weather patterns (e.g., Hewitson and Crane 2002; Cassano et al. 2006; Gibson et al. 2017; Nygård et al. 2019). In this study, we allocate 20 characteristic atmospheric circulation types in spring (MAM, 2007–2018), using mean sea-level pressure (MSLP) data of ERA5 reanalysis (Copernicus Climate Change Service, 2017) produced by the European Centre for Medium-Range Weather Forecasts at 6 h interval as the input data. MSLP is used here as it is a robust indicator of the atmospheric state in the Arctic, and captures and represents the circulation and flow patterns that affect the lower troposphere (Neal et al., 2016 and the references therein). This is important for analyzing the pollution transport processes that occur mostly in the lower troposphere and their subsequent impacts. In the initial phase of the SOM analysis, each of the 20 nodes in the SOM array has an associated reference vector with an equal dimension to the input MSLP data. Then, each time step of input MSLP data is compared with the reference vector of each node during the SOM training. The reference vectors, which are most similar to the input data vector, are adjusted towards the input data vector. This procedure is repeated until the reference vectors do not change anymore. The subsequent output is a two-dimensional SOM array of gridded MSLP fields, having probability density of the input MSLP data. This array is organized according to similarities in CTs, having the most similar circulation patterns located next to each other and the most dissimilar patterns in the corners of the array. Each time step of the input MSLP data is linked to the most similar circulation type or weather state (node) in the array. Based on these time steps, we are also able to form composites of trace gases in each CT separately. Although we originally investigated 20 CTs, we present here the results for 8 of those 20 CTs

for the sake of brevity. The selection of these 8 CTs is based on a) the strength of the signal observed in the trace gases, 2) the frequency of occurrence of the circulation types, and 3) the diversity of the CTs. The results for all 20 CTs are kept in Appendix-1 for interested readers. The 8 CTs (CT1 - CT8) selected in this study (shown in Fig. 2) correspond respectively to CT1, CT4, CT9, CT12, CT14, CT18, CT19 and CT20 in Fig. A1 in the Appendix-1.

After deriving the prevalent circulation types, the climatological means of $NO_2$, CO, $O_3$ and AOD during the March, April and May months are computed separately. For each circulation type, the number of days that represent that type could be different in each month. In order to compute a climatological mean that takes into account this difference, we weighed the climatological means of each month with weighing factors shown in Fig. 1, giving climatological means of $NO_2$, CO, $O_3$ and AOD associated with each weather state. The composites of $NO_2$, CO, $O_3$ and AOD for each weather state are then computed. The anomalies shown later are the differences between these composites and the weighted climatological means for each weather state. Only those trace-gas anomalies that are statistically significant using Student's t test at 90% confidence interval are shown.

## 3 Overview of the CTs and associated meteorological conditions

Fig. 2 shows the mean MSLP patterns during the 8 CTs that are chosen from the SOM analysis. These types are mainly characterized by different locations and strengths of cyclones and anticyclones with respect to one another. For example, the first CT (CT1) is characterized by the most commonly observed low pressure regimes in the Northeast Atlantic and European Arctic and an intense Beaufort high on the Pacific side of the Arctic. In CT2, the low pressure systems in the Greenland and Norwegian Seas gradually intensify, while the anticyclones move over the Chukchi and East Siberian seas and weaken in their intensity. In the case of CT3, almost half of the Arctic (Greenland, Canadian archipelago and Beaufort Sea and Alaska) is under the influence of a strong anticyclone, with the center of action located east-west of the international date line. The strongest anticyclonic conditions are observed during CT3, while the strongest cyclonic conditions are observed in CT2 over the Greenland and Norwegian Seas. CT4 shows a tripole pattern wherein strong low pressure systems are located over the Barents and Kara seas as well as over Alaska at the opposite side of the Arctic, while a weaker but noticeable high pressure zone is observed over the Canadian archipelago. CT5 is characterized by anticyclonic conditions dominating over the entire central Arctic as well as Greenland and the Canadian archipelago. CT6 and CT7 show dipole patterns (with different intensities) with cyclonic conditions over Siberia and anticyclonic conditions prevailing at the opposite side of the Arctic circle. Finally in CT8, cyclonic conditions are observed both in the Northeast Atlantic and Siberia, while the anticyclonic conditions over Scandinavia and Beaufort Sea. The SOM analysis presented in Fig 2 reveals how varied and complex the atmospheric large scale circulation patterns can be over the Arctic in spring.

Atmospheric circulation drives the transport in the atmosphere. For example, it largely distributes moisture in the Arctic atmosphere by dictating its horizontal transport and modulating the local evaporation at the surface. Fig. 3 shows the specific humidity anomalies (dq), based on AIRS data, associated with those 8 CTs. These anomalies are a good indicator (and manifestation) of the transport patterns shaped by the cyclonic and anticyclonic conditions mentioned above. Furthermore, atmospheric humidity has an impact on the aerosol optical properties and morphology as well as on the processes affecting the lifetime of trace gases. An increase in dq is detected in the Greenland and Norwegian Seas in CT1 and CT2 due to the cyclonic conditions in the Northeast Atlantic transporting more heat and moisture. In CT3, in the absence of such transport in the Northeast Atlantic and due to the presence of anticyclones over Greenland and Canadian archipelago, drier and cooler air masses are transported over the Greenland, Norwegian and Barents Seas, as seen in the reduction of humidity anomalies over these areas. A similar decrease in humidity is also observed in CT4 and CT6 over Greenland and Norwegian Seas. In CT4, an increase in dq can be seen over the Laptev Sea as a result of the strong low pressure systems centered eastward of Scandinavia over the Barents and Kara Seas along the Russian coast. In CT8, a slight increase in humidity is seen over the entire Arctic Ocean due to the influence of low pressure systems located over the Northeast Atlantic and northern Siberia. Our results indicate that the CTs derived based on MSLP can also be used to analyze the free and upper tropospheric pollutants. The AIRS derived geopotential height anomalies at 500 hPa are shown in Fig. 4. There is a coupling between the lower and upper level circulation during those circulation types and, especially, a good resemblance in the locations of the centers of action of low pressure systems and anticyclones derived based on MSLP and the 500 hPa geopotential heights.

## 4 Covariability of CTs and air pollutants

The response of $NO_2$ to the CTs is shown in Fig. 5 in terms of weighed anomalies. It is to be noted that, while the SOM analysis is done over the region northward of 60N in order to emphasize the circulation patterns in the Arctic region, we present the anomalies of the pollutants northward of 50N in order to provide the large-scale spatial context. It can be seen that the spatial distribution of $NO_2$ is highly sensitive to the CTs, not only over the polluted mainland and source regions, but also over the Arctic Ocean. Particularly over northern Europe, a distinct pattern emerges, wherein the $NO_2$ anomalies change sign gradually from CT1 to CT8 in response to the changing atmospheric circulation patterns. In CT1 and CT2, there is a clear transport signal in the $NO_2$ anomalies. The location of low pressure systems in the Northeast Atlantic favors the transport of $NO_2$ from the northern, central and eastern European regions into the Arctic. The increased specific humidity anomalies in the European Arctic further confirm such a transport (Fig. 3). The strongest signal is observed in CT2, when the center of action of polar vortex is located over Greenland (Figs. 2 and 4) and the intensity of the vortex is stronger, favouring the increased transport of $NO_2$ in the Barents and Kara Seas, reaching even further north into the Arctic. Previous studies have indicated that, in the European sector of the Arctic, such transport occurs predominantly in the lower troposphere

(Stohl, 2006; Thomas et al., 2019). A pronounced increase in humidity anomalies is also seen over these regions in CT2. Among all circulation types, the highest $NO_2$ anomalies are observed over Scandinavia in CT1 and CT2, suggesting a noticeable influence of these circulation types in the pollution variability in these countries. The transport from the central and eastern European countries is especially prominent in CT2. It is to be noted that the circulation types, CT1 and CT2, roughly resemble the typical loading patterns of North Atlantic Oscillation and/or Arctic Oscillations over the central and Eurasian Arctic, which is shown to have a noticeable impact on the pollutant variability over these regions (e.g. Eckhardt et al., 2003; Christoudias et al., 2012).

An entirely opposite $NO_2$ response is seen in CT5 to CT8. In CT5 and CT6 with anticyclonic conditions prevailing over Greenland and northern north Atlantic at varying intensity, the transport of cleaner air masses from the central Arctic lead to negative $NO_2$ anomalies over the central and northern parts of Europe. The anticyclone further moves eastwards over Greenland and Norwegian Seas and over northern Scandinavia from CT5 to CT6, blocking the transport from the southerly latitudes and therefore leading to negative $NO_2$ anomalies during these circulation types. In CT7, the circulation pattern in the Canadian archipelago and European sector of the Arctic, together with cyclonic conditions in central and Eastern Siberian regions facilitate the north east Asian transport of $NO_2$ into Alaska and northern Canada. In CT8, the low pressure systems over the Northeast Atlantic and Siberia, Kara and Laptev Seas lead to a slight increase in $NO_2$ concentration in the Barents Sea. The blocking over southern Scandinavia and Europe however limits the large-scale transport into the Arctic from the European sector. The northeast Asian regions and northern Pacific Ocean show no sensitivity to the circulation types, most likely due to the persistent nature of westerly winds over this region in combination with the persistent continental pollution outflow over the northern Pacific.

The $O_3$ anomalies at 925 hPa also show sensitivity to the circulation types. They appear to be opposite in nature to that of the $NO_2$ anomalies. For example, a reduction in the $O_3$ concentrations over northeast Atlantic and Scandinavia seen in CT1 and CT2 is consistent with the strong $NO_2$ increases observed during the same circulation types. A statistically significant increase in the central Arctic is seen in CT1 and CT3. However, the corresponding $NO_2$ anomalies over the central Arctic in these circulation types are not statistically significant. An inverse correspondence between $O_3$ and $NO_2$ away from the source regions is not expected due to the different life times, aging and transport processes. A decrease in $O_3$ concentrations over the central Arctic corresponds to the presence of cyclonic conditions over Eurasia and Siberia (CT6-CT8).

The springtime photochemistry in the Arctic is very complex, as duly noted in the rich literature that documents the research and observations on this subject matter (Lu et al., 2019 and the references therein). The interactions between $NO_2$ and $O_3$ are also highly non-linear in reality and hence a one-one correlation can not be established. In the troposphere, NO is converted to $NO_2$ in the presence of $O_3$ which is a potential sink for $O_3$. However, during sun-lit conditions, $NO_2$ is converted back to NO via photolysis which results in $O_3$ production. Apart from these chemical reactions, local meteorological conditions such

as temperature, relative humidity and rainfall play an important role in the production and dispersion of these pollutants. Stratospheric intrusions are another source of $O_3$ variability in the troposphere that may play a role under different circulation types (Yates et al., 2013; Langford et al., 2015; Lin et al., 2015). The persistent anticyclonic conditions could not only lead to the accumulation of the tropospheric $O_3$, but also favour the large-scale descent or intrusions into the lower troposphere, leading to positive $O_3$ anomalies.

Fig. 7 shows the tropospheric AOD anomalies based on the CALIOP-CALIPSO aerosol profile product. It is to be noted that, being an active profiler, the spatial coverage of CALIOP-CALIPSO is very poor and the anomalies look patchy, particularly over the inland regions because of a limited number of samples for each circulation type. The passive imagers either do not have AOD data available in spring (due to poor illumination conditions) or the quality of the retrievals can be very poor due to the challenging surface conditions and the underlying uncertainties in cloud masking. CALIOP provides the most accurate sampling of aerosols over the Arctic Ocean in spring in comparison to passive imagers, but with this trade-off of having poor spatial sampling and therefore the AOD data have to be interpreted cautiously. We, nonetheless, decided to include CALIPSO in the analysis since it can provide an important context while studying the trace gas variability. For example, we can see that there are at least two signals that are robust and consistent with other observations. An increase in AOD in CT1 and CT2 is observed in Greenland and Norwegian Seas and northern Scandinavia, which is consistent with the increases in $NO_2$, further confirming the role of these circulation types in transporting the pollutants into the Arctic. An increase in humidity, as mentioned earlier, in CT1 and CT2 impacts the AODs due to increased water uptake during transport. These circulation types are similar to those that could change the stability regimes as a result of heat and moisture transport over the colder sea-ice surfaces in the inner Arctic and trapping the aerosols and pollutants below the inversions in the Eurasian sector of the Arctic, as previously reported in Thomas et al., 2019. The opposite tendencies in CT5 and CT6, wherein the negative AOD anomalies are observed over the Norwegian Sea and northern Scandinavia, are also consistent with the $NO_2$ decreases observed in these circulation types. The anticyclones prevailing over Greenland and north of Scandinavia block the transport of trace gases and aerosols into the Arctic during these circulation patterns. The increased AODs along the western coast of Scandinavia in CT3 could be due to the location of anticyclone in the Arctic and the low pressure systems in central Europe that transport pollutants from the eastern Europe and western parts of Russia, including the biomass burning regions, over these coastal regions. In the case of other circulation types, the AOD anomalies are too patchy to draw meaningful conclusions in the sense that there are no consistent features either with the meteorological conditions or other pollutants.

Unlike tropospheric $O_3$ and $NO_2$, CO has an atmospheric lifetime ranging from few weeks to few months and therefore is often regarded as a suitable tracer to study the long-range pollution transport. Due to its longevity, the spatial distribution of CO in the free troposphere is also quite homogeneous compared to other trace gases and the local pollution variability is

often diffused in the large-scale signal. However, CO is an excellent tracer to study the coupling between the pollution variability in the free troposphere and the lower tropospheric circulation patterns, given the influence of these CTs on the entire troposphere, and also to study the large-scale, first order impact of the CTs on the free tropospheric pollutants. Such a large-scale signal is indeed visible in the CO anomalies shown in Fig. 8. Two main regimes can be seen; one dominated by the Arctic-wide increases in the CO concentrations (eg. CT1 to CT2) when the low pressure systems are active in the North Atlantic and the other when the decreases in the CO concentrations (eg. CT5 to CT7) can be seen over much of the Arctic likely due to the atmospheric blocking over those regions. The CO anomalies over Scandinavia, northeast Atlantic, Greenland, Norwegian and Barents Seas show the strongest sensitivity to the circulation types.

## 5. Conclusions

The transport and the distribution of the pollutants in the Arctic, especially that of the short-lived climate forcers, depends heavily on the prevailing atmospheric circulation patterns. Understanding pollutant variability in relation to the dominant circulation types is therefore important. Here, we investigate the concentrations of $NO_2$, $O_3$, CO and aerosols and their co-variability during the 8 different circulation types in the spring season (March, April and May) over the Arctic. The circulation types discussed in this study are derived by the Self-Organizing Maps analysis of MSLP. A combination of satellite based and reanalysis datasets spanning over 12 years (2007-2018) is used. The following conclusions are drawn from the analysis.

1. The 8 characteristic circulation patterns during spring, allocated by the SOM analysis based on the MSLP fields, represent different locations and intensities of cyclonic and anticyclonic events in relation to each other. The MSLP circulation patterns are connected to 500 hPa geopotential height anomalies and also shape the atmospheric humidity distribution. The circulation patterns largely dictate the transport in the atmosphere, especially from the main source areas in the southerly latitudes into the Arctic.

2. It is observed that all pollutants investigated here show sensitivity to the circulation types and some common patterns emerge in their response. $NO_2$ shows the strongest sensitivity among the trace gases and aerosols analyzed here.

3. The circulation types (CT1, CT2) with low-pressure systems located in the northeast Atlantic show a clear statistically significant enhancement of $NO_2$ and AOD in the European Arctic. The $O_3$ concentrations are however decreased in such events.

4. The circulation types (CT5, CT6, and CT7) with atmospheric blocking over Greenland and northern Scandinavia show the opposite signal, in that the $NO_2$ concentrations are decreased and AODs are smaller than the climatological values. The $O_3$ concentrations are however increased during these events in the European Arctic.

5. The first order signal of the influence of circulation types on the free tropospheric CO is seen, with two main regimes emerging. The first regime shows the Arctic-wide positive anomalies in the CO concentrations when the low pressure systems are active in the North Atlantic and the other when the negative CO anomalies are observed due to the atmospheric blocking over those regions.

The present study provides the most comprehensive investigations so far of the sensitivity of springtime pollutant distribution to the atmospheric circulation types in the Arctic, also providing an observational basis for the evaluation of chemistry transport models.

**Acknowledgements**

The study was funded by the Swedish National Space Agency (grant number 94/16). TN acknowledges the funding by the Academy of Finland via project TODAy (grant number 308441). The authors would like to thank the OMI, AIRS and CALIPSO Science Teams for the data products as well as CAMS and ECMWF for the corresponding reanalysis data products.

**Code/Data availability**

All datasets used in the present study are publicly available below:
https://disc.gsfc.nasa.gov/datasets/OMNO2G_003/summary

https://airs.jpl.nasa.gov/data/get-data/standard-data/

https://atmosphere.copernicus.eu/data

**Competing interests**

The authors declare no competing interests.

**Author contributions**

MT and AD designed the study. MT carried out the analysis and wrote the first draft of the manuscript. TN performed and provided the SOM analysis. All authors contributed to the writing and interpretation of the results.

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

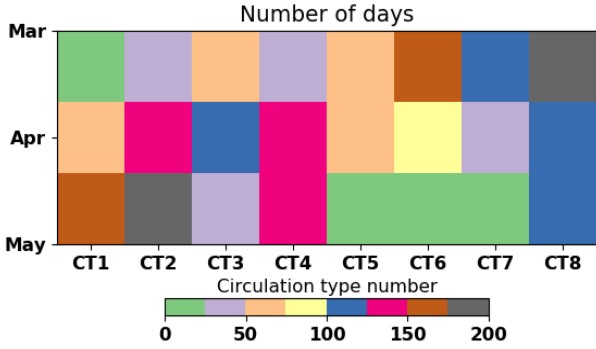

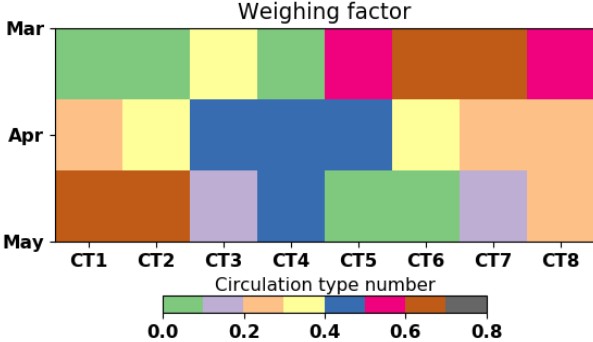

**Fig. 1: (top) The number of days analysed for each circulation type and month and (bottom) the corresponding weighing factor used to compute the climatologies of trace gases and aerosols.**

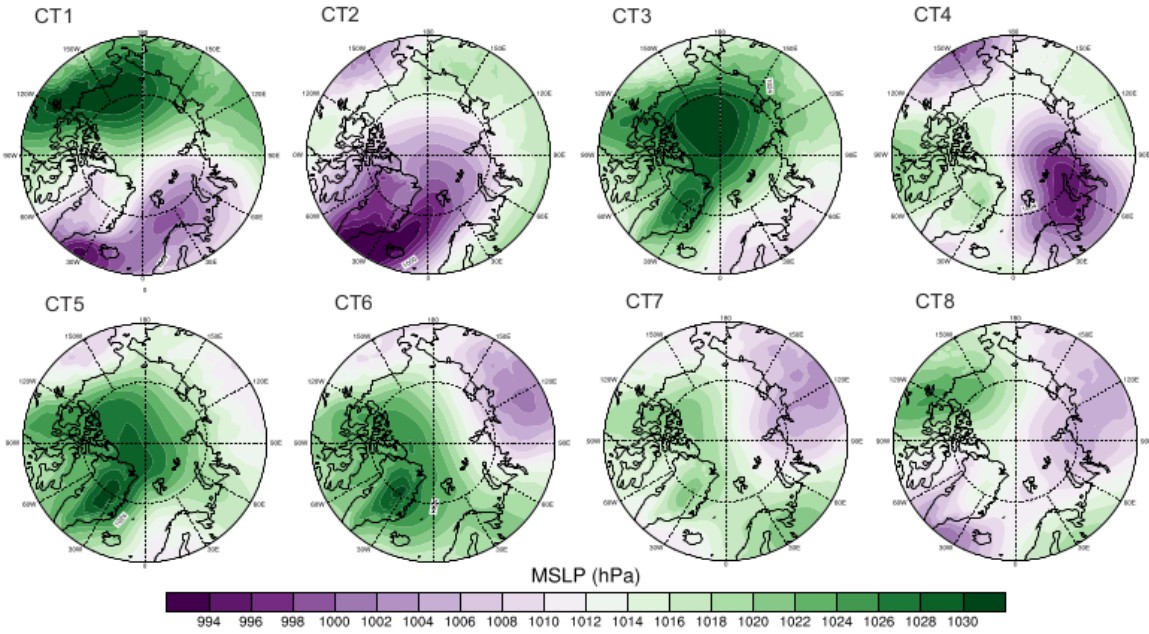

**Fig. 2: Mean sea level pressure (MSLP) averaged over the cases belonging to each of the 8 circulation types chosen from the 20 circulation types shown in the Appendix-1.**

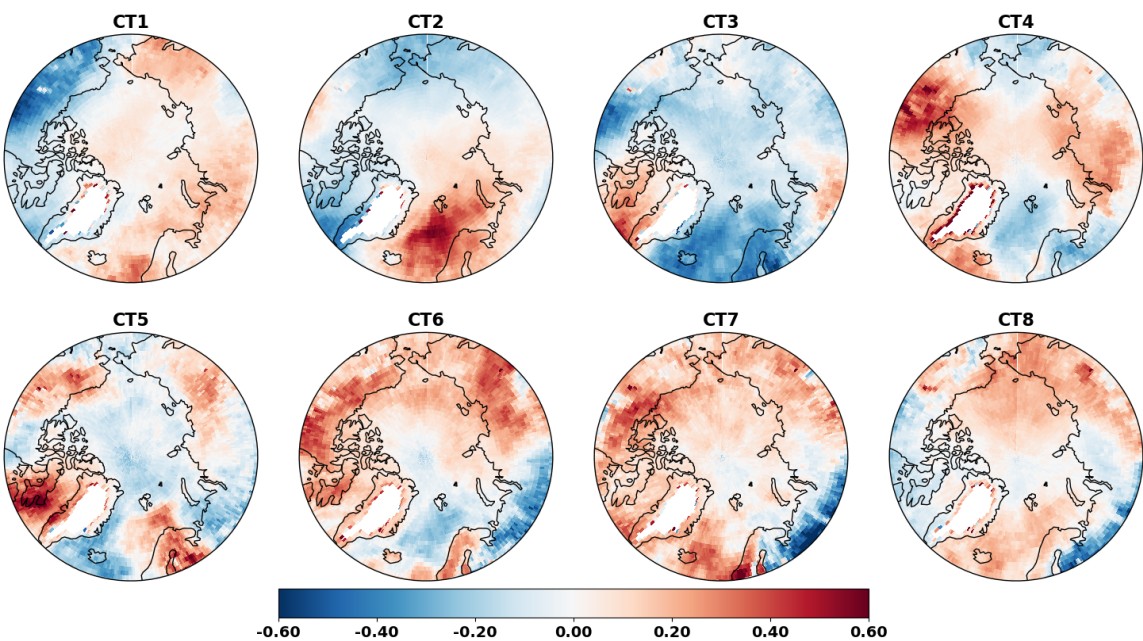

**Fig. 3: 850 hPa specific humidity anomalies (g kg⁻¹) based on AIRS data for the 8 circulation types.**

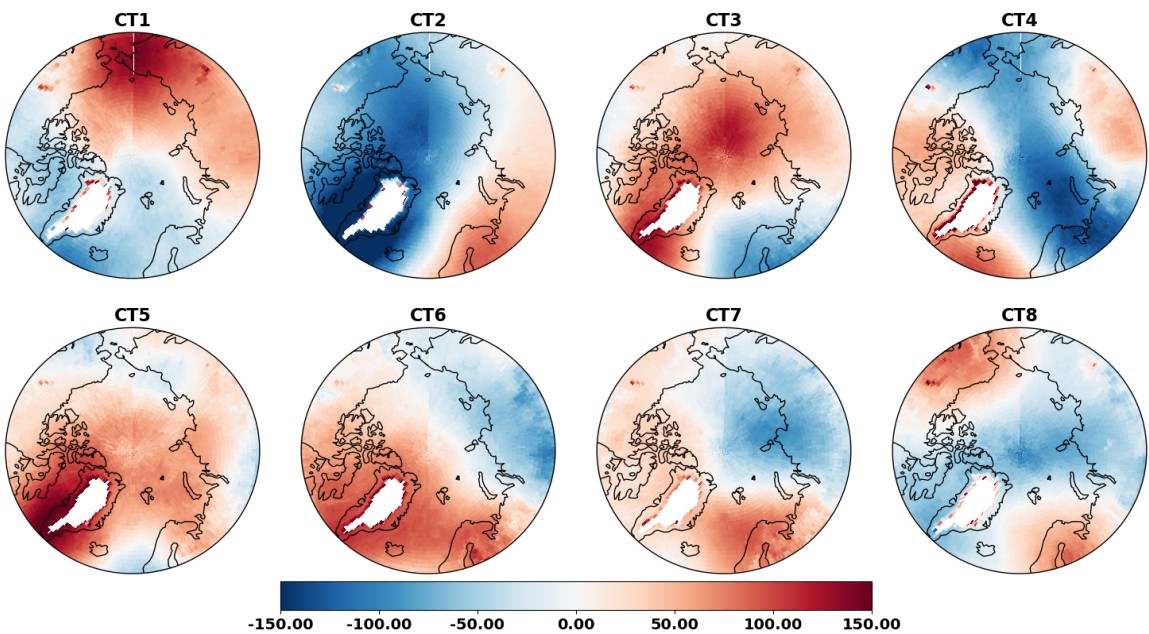

**Fig. 4: Geopotential height anomalies (m) at 500 hPa based on AIRS data for the 8 circulation types**

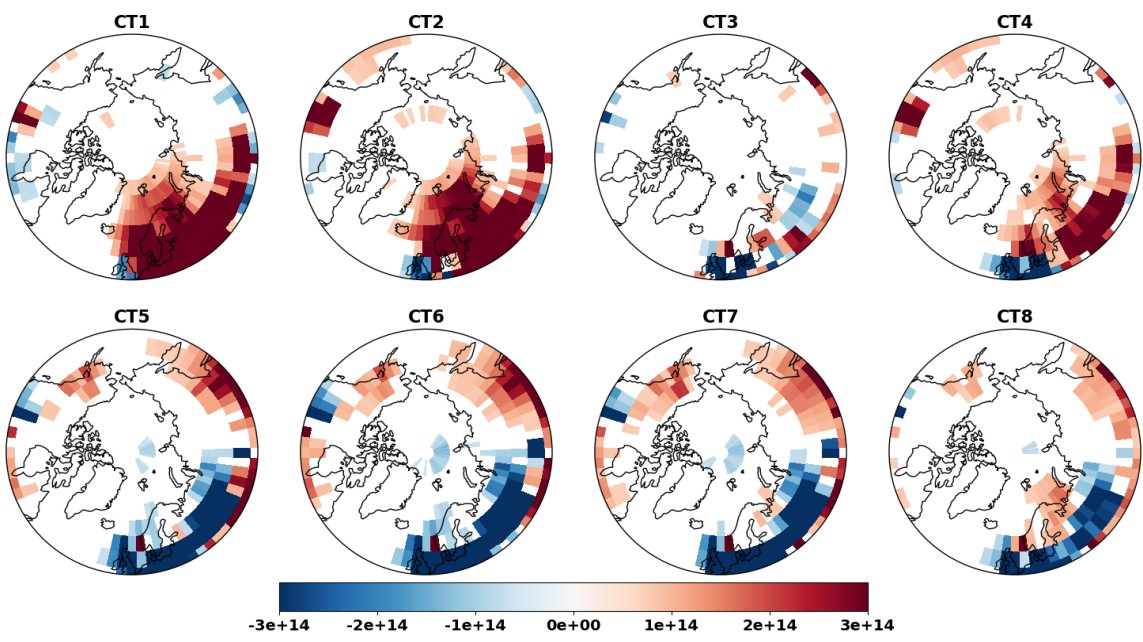

**Fig. 5: NO$_2$ total column anomalies (molecules/cm$^2$) based on OMI data for the 8 circulation types. Only those anomalies that are statistically significant at 90% confidence are shown.**

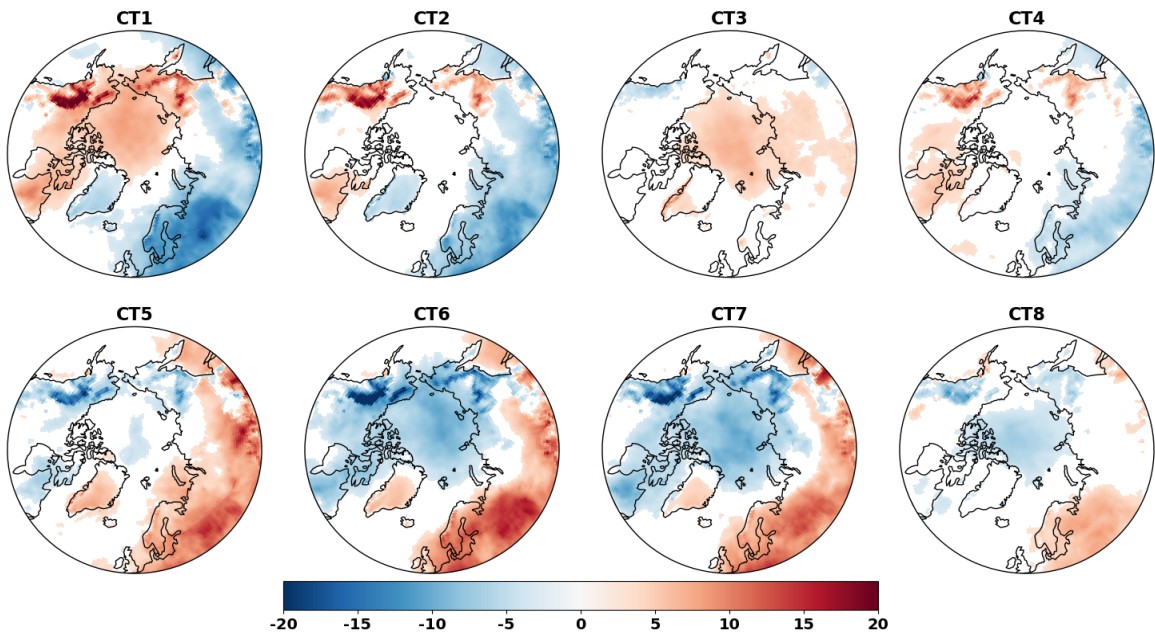

**Fig. 6:** 925 hPa $O_3$ anomalies (ppbv) based on CAMS data for the 8 circulation types. Only those anomalies that are statistically significant at 90% confidence are shown.

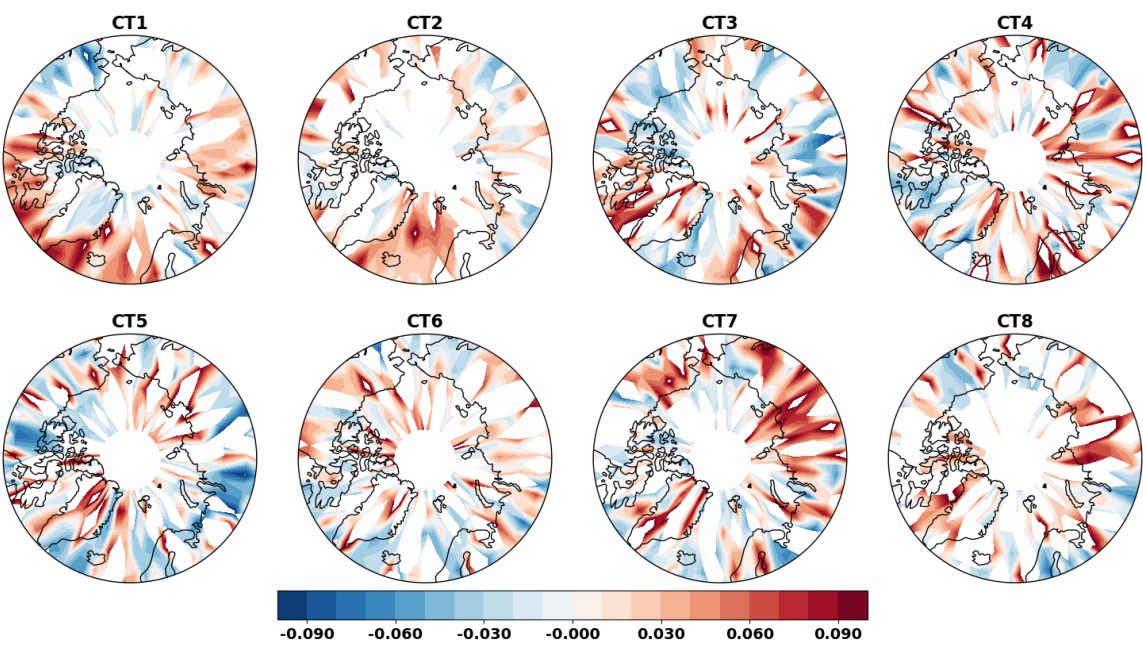

**Fig. 7:** Tropospheric AOD anomalies based on CALIOP-CALIPSO data for the 8 circulation types with 90% confidence.

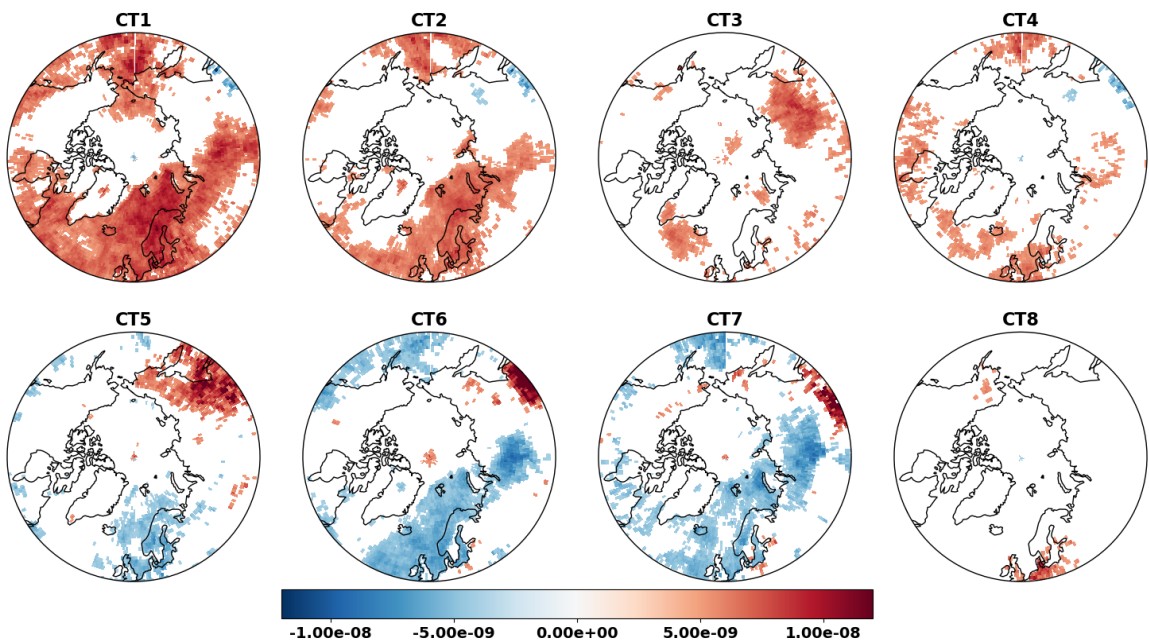

**Fig. 8: 500 hPa CO anomalies as volume mixing ratios based on AIRS data for the 8 circulation types with 90% confidence.**

**Appendix-1**

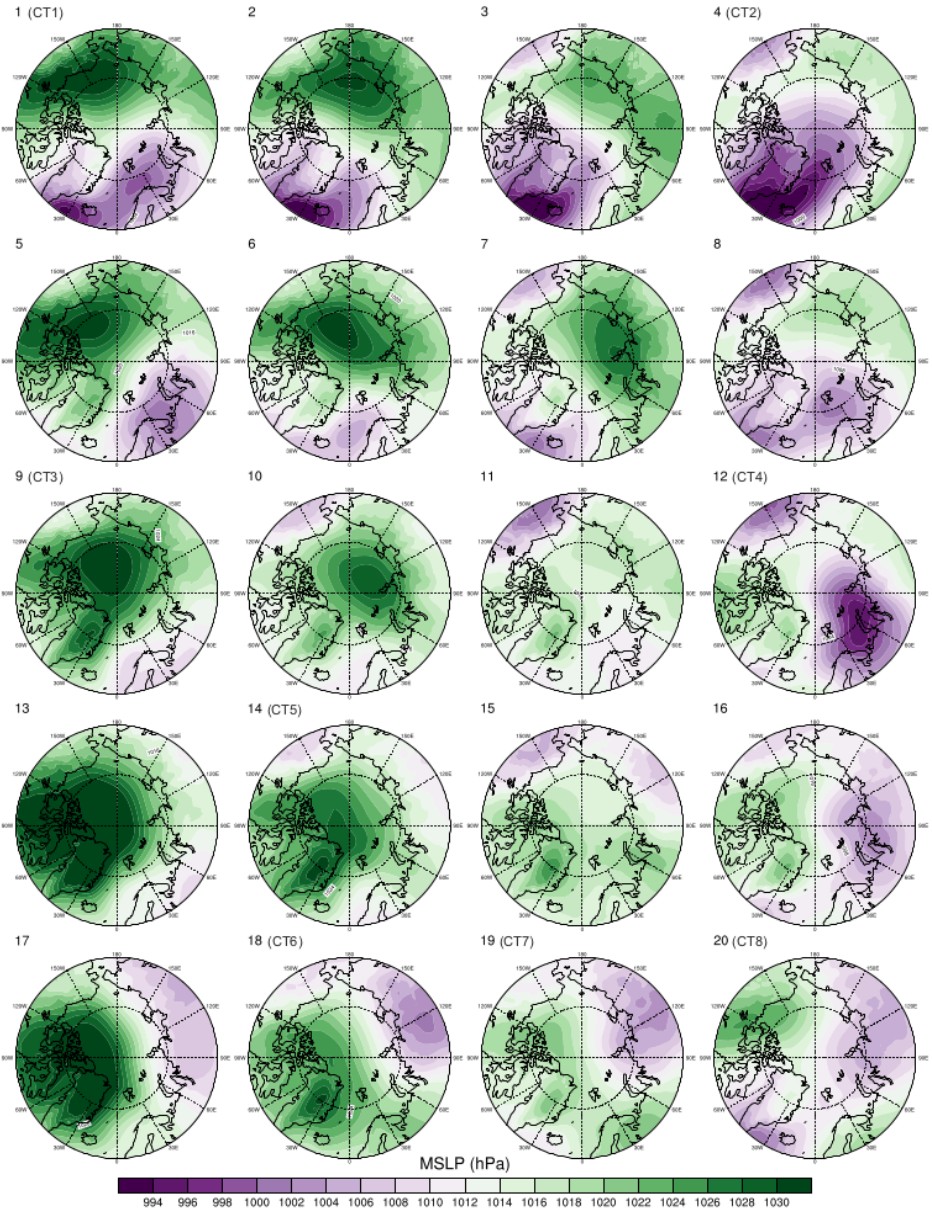

**Figure A1: Mean sea level pressure (MSLP) averaged over the cases belonging to each of the 20 circulation types. The chosen 8 circulation types are shown in the brackets.**

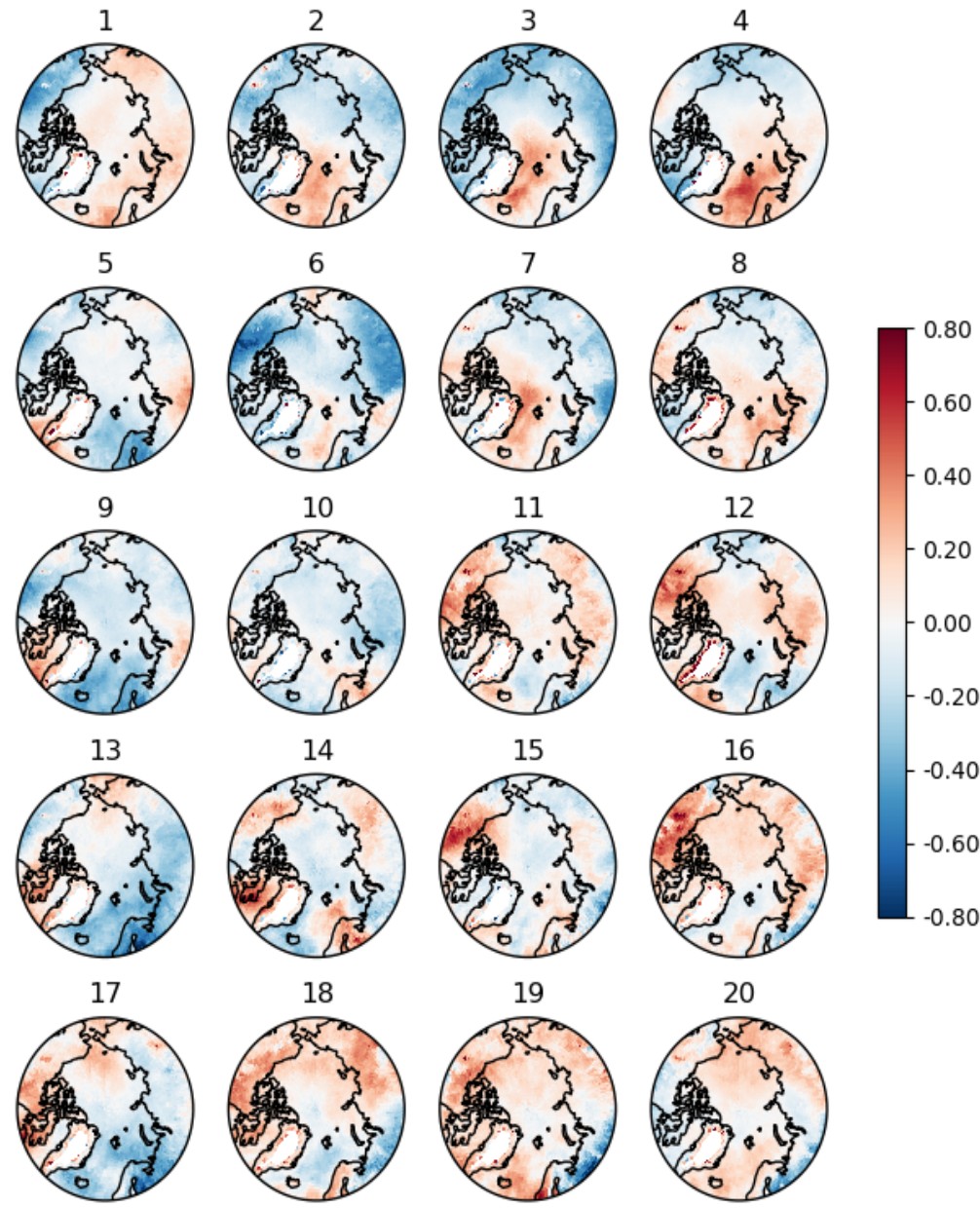

**Figure A2: Specific humidity anomalies (g/kg) at 850 hPa based on AIRS data for the 20 circulation types.**

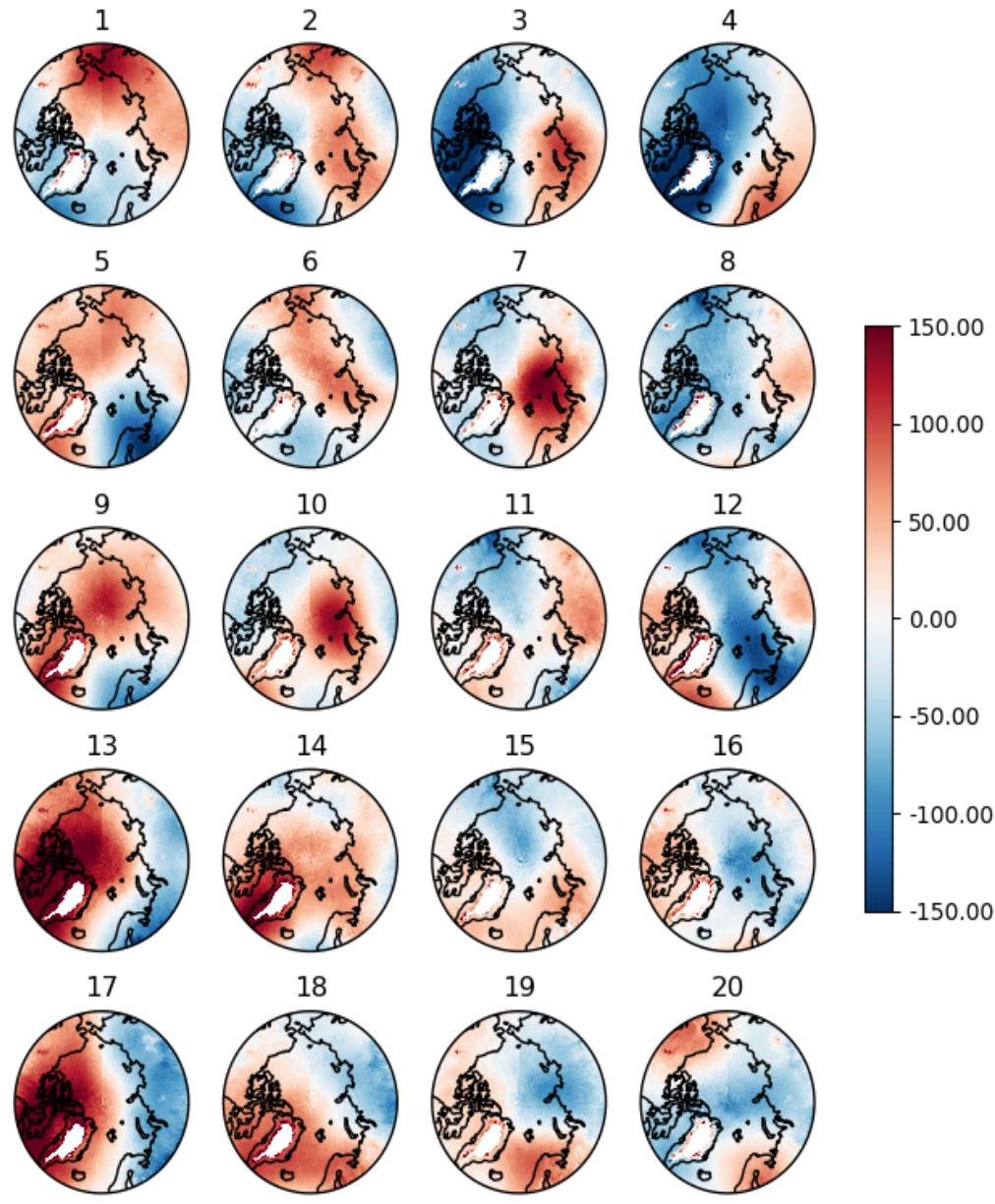

**Figure A3: Geopotential height anomalies (m) at 500 hPa based on AIRS data for the 20 circulation types.**

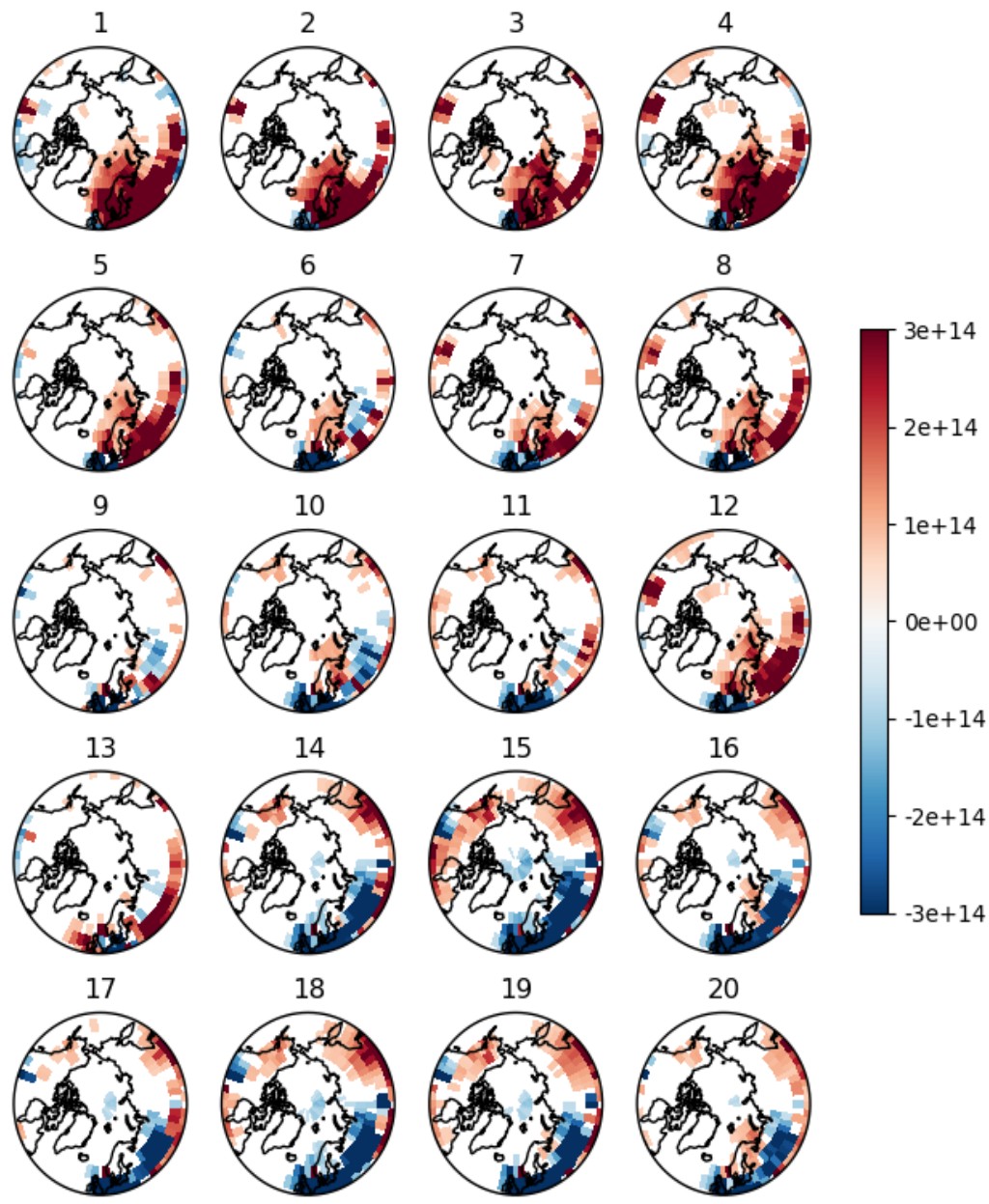

**Figure A4: NO₂ total column anomalies (molecules/cm²) based on OMI data for the 20 circulation types. Only those anomalies that are statistically significant at 90% confidence are shown.**

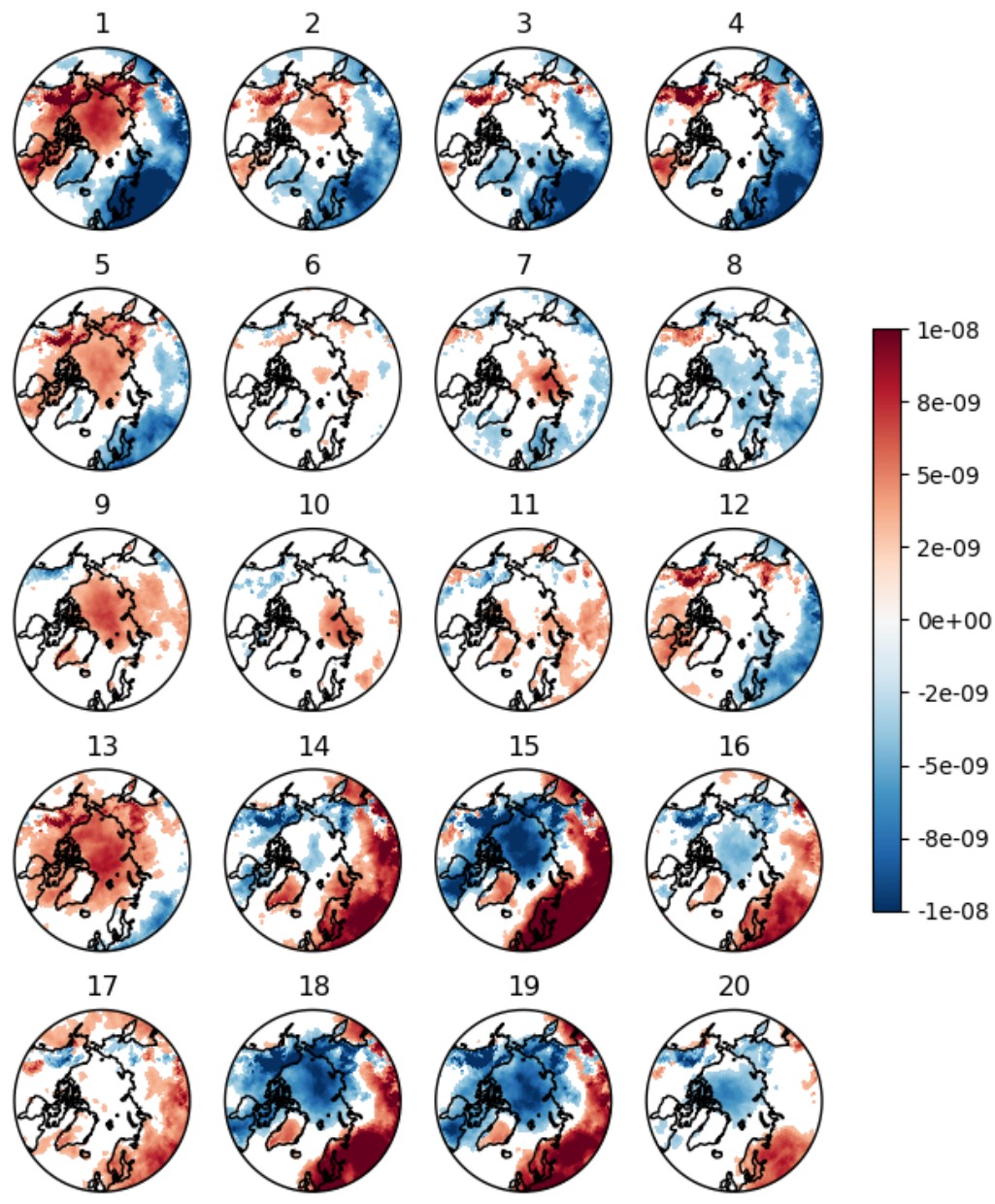

**Figure A5: 925 hPa O₃ anomalies as volume mixing ratios based on CAMS data for the 20 circulation types. Only those anomalies that are statistically significant at 90% confidence are shown.**

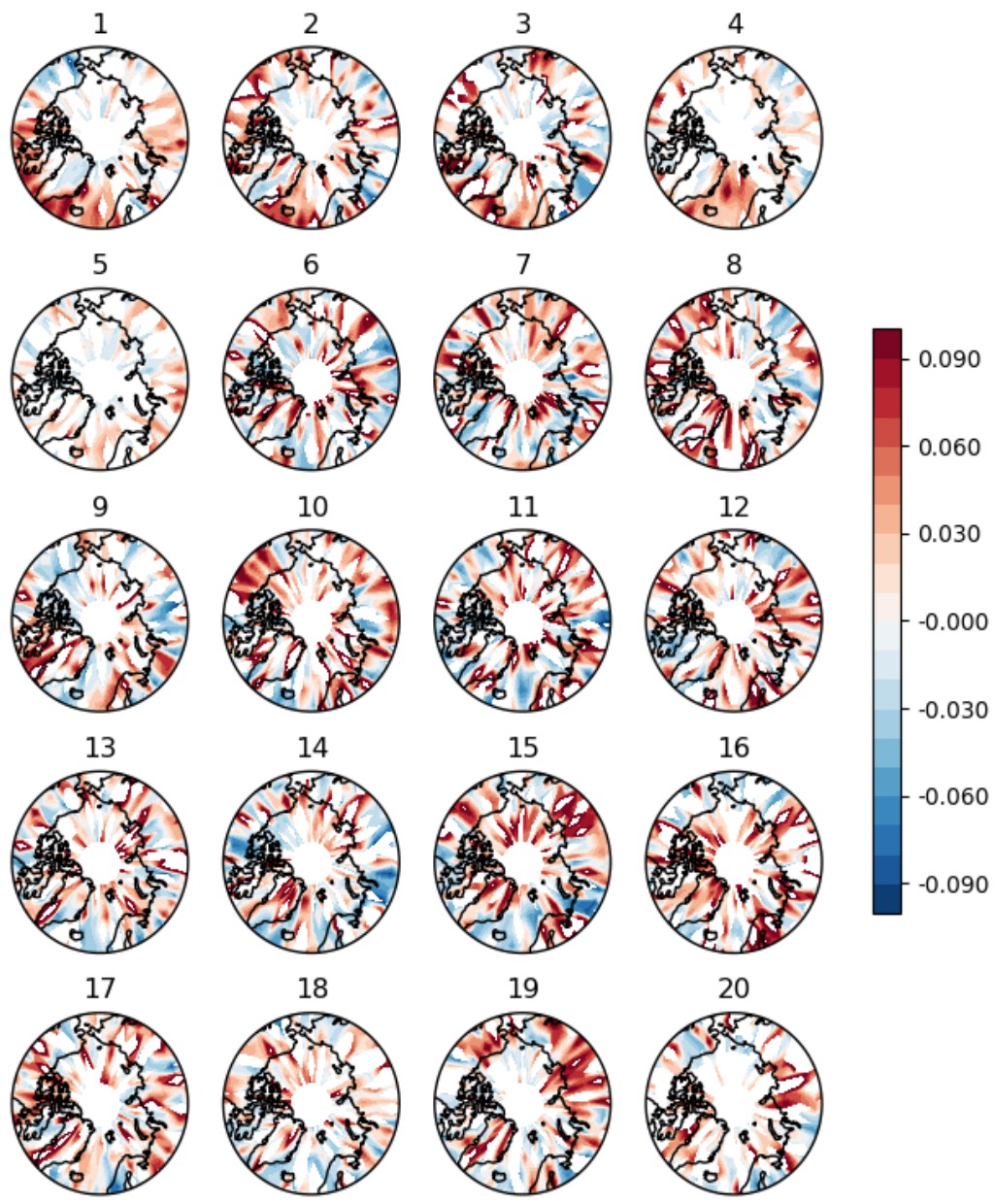

**Figure A6: Tropospheric AOD anomalies based on CALIOP-CALIPSO data for the 20 circulation types. Only those anomalies that are statistically significant at 90% confidence are shown.**

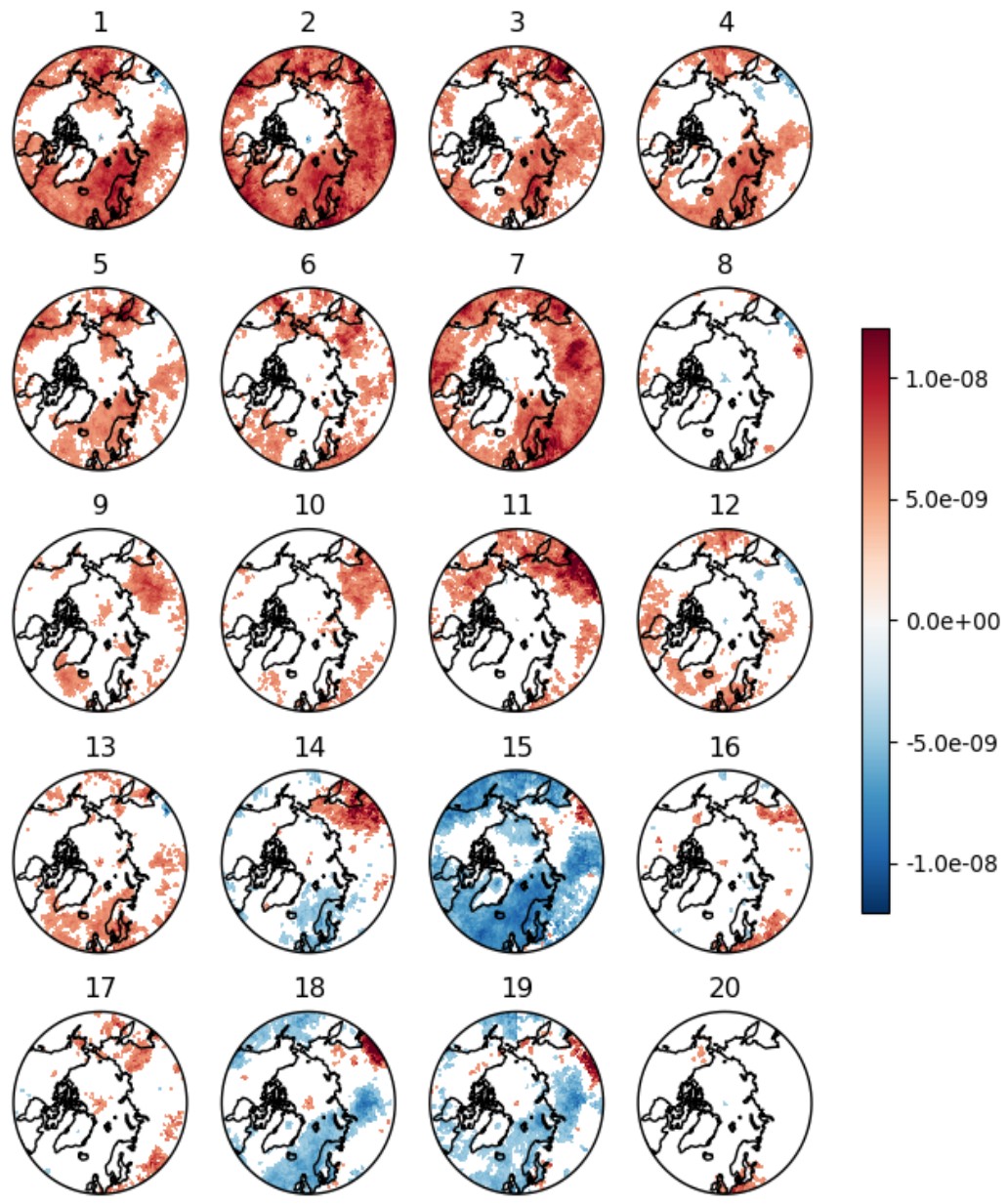

**Figure A7: 500 hPa CO anomalies as volume mixing ratios based on AIRS data for the 20 circulation types. Only those anomalies that are statistically significant at 90% confidence are shown.**