# Peer review of "Influence of springtime atmospheric circulation types on the distribution of air pollutants in the Arctic"

_Atmospheric Chemistry and Physics, 2021_

## Author Comment (AC1)

**Response to Referee #1**

This paper applies Self-Organizing Maps to sea level pressure fields to identify 20 circulation patterns in the Arctic spring, and then analyzes the observed distributions of pollutants associated with these patterns. The analysis aims to demonstrate how the transport and distribution of pollutants in the Arctic varies depending on the circulation pattern and to provide an observation-based test of chemistry transport models. This is an original and interesting idea, and the Self-Organizing Map method is state-of-the-art. However, more discussion of uncertainties and sampling of the satellite data in the Arctic is needed. In addition, the inclusion of 20 different circulation patterns makes the results complicated to interpret. I list general and specific comments below.

We thank the referee for the encouraging words and constructive suggestions. Please find below point by point reply to your comments.

**General comments:**

1. The presence of snow and ice, as well as cloud cover, can pose challenges for satellite retrievals or affect how much data is available, potentially leading to sampling biases. Please include more discussion of the sampling and any uncertainties for each satellite product in the Arctic environment. If ground or aircraft-based observations are available to validate the findings, that would also strengthen the paper.

We agree completely with the referee that the surface conditions and cloud cover play an important role in the data sampling. We have, in fact, examined these issues in detail in the beginning when the experimental setup was designed, based on the experience of using these OMI, AIRS and CALIPSO datasets in the high latitude regions.

Different considerations are required for different satellite sensors and the trace gas in question.

For example, please find below the results from a sensitivity study, which shows the comparison of NO2 anomalies during the first four circulation types in two scenarios, namely a) when the all-sky OMI NO2 retrievals are analysed (top row) and b) when the stricter cloud clearing criteria is used in that the retrievals are considered only when the total cloud cover is less than 30% (bottom row). There are undoubtedly some small regional differences in the anomalies in both cases, but the overall signal is very robust. We clearly see the transport of NO2 in the Arctic in both cases.

[Figure]

In the case of carbon monoxide from AIRS, we have allowed retrievals when the cloud cover is up to 70%. This is mainly because the hyperspectral capability of AIRS allows relatively accurate retrievals even under the presence of partial cloudiness. Moreover, the high latitude regions are often characterized by the presence of either low level boundary layer clouds or the high thin cirrus clouds, both of which do not significantly affect the AIRS retrievals in the free troposphere at 500 hPa. Below we show the results from another sensitivity study wherein CO anomalies for the first circulation type are shown when the AIRS cloud fraction is constrained to 30%, 50% and 70%. Here as well, we can see that there are some small regional differences, but the main signal remains robust.

[Figure]

The aerosol retrievals from CALIOP-CALIPSO are surface blind and CALIOP is probably the best sensor to date to delineate aerosols from clouds. We have used CALIOP retrievals only when the

Cloud-Aerosol Discrimination (CAD) Score is between (and equal to) -100 and -20, thereby ensuring that the selected features are indeed aerosols. We have furthermore used retrievals only when the CALIOP extinction quality flag is 0, 1 or 2, ensuring the successful retrievals.

It is also worth pointing out that previous studies have shown that the circulation patterns that favour pollution transport into the Arctic are also associated with the transport of heat and moisture into the Arctic, which in turn leads to increased cloudiness (Devasthale et al., 2020; Thomas et al., 2019; Johansson et al., 2017). Therefore, we decided to relax the cloud clearing thresholds in order to capture these most realistic scenarios (while ensuring that the broader signal is not affected by such relaxation). By imposing a strict threshold on cloud cover (for example, analysing only clear-sky conditions to ensure the best quality retrievals) would introduce unrealistic clear-sky biases in the anomalies shown in the manuscript.

We have added a discussion regarding this point in the revised manuscript.

2. Section 2 mentions that ozone at 925 hPa from CAMS is used in the analysis because of the lack of reliable lower tropospheric ozone observations. Does this mean that the CAMS ozone at this level is primarily model-based? Has it been validated for the Arctic? This should be discussed since it is relevant to whether this method provides an observation-based test of chemical transport models.

The following sentences have been added to the manuscript "The validation of the ozone CAMS reanalysis product is carried out extensively using ground based measurements (TOAR database for surface ozone (Schulz et al., 2017a; 2017b) and ozonesondes globally (Inness., et al., 2019; Huijnen, et al., 2020). CAMS assimilation system makes use of data from SCIAMACHY, MIPAS, OMI, MLS, GOME-2, and SBUV/2 for ozone. Even though the surface ozone is primarily model based, upgrades in the CAMS chemical data assimilation system, assimilated measurements etc have improved the near surface estimates.".

3. What is the reason for allocating 20 circulation types? Could this number be reduced? The discussion often refers to multiple types together. For example, line 183 mentions 4 types under the influence of a strong anticyclone. Are these 4 still completely different patterns? It is also difficult to intuitively visualize the distinction between the 20 different maps presented in the plots, as the same main features seem to be present in multiple maps. If the number of maps were reduced, or perhaps the presentation of the plots organized to focus on a smaller number of clearly-distinguishable ones, the discussion would be easier to follow.

We originally wanted to capture as many different circulation types as possible. Having 20 circulation types means that some of them may be similar over certain regions or may not show a strong signal. We nonetheless decided to include them all to avoid doing some sort of "cherry picking". We do however agree that it is possible to reduce them and achieve a balance. The Referee #2 also raised a similar issue.

In the revised manuscript, we have therefore included only 8 circulation types. The selection was based on a) the strength of the signal observed in the trace gases 2) the frequency of occurrence of the circulation types and 3) the diversity and strength of the circulation pattern. The results for all 20 circulation types will be kept as the Supplementary Material.

4. One suggestion for presenting the main results more clearly is to include a figure that shows all of the pollutant anomalies (CO, NO2, O3, AOD) side by side for a couple of the main circulation patterns, so that the reader can easily see how anomalies in different pollutants relate to each other for a given circulation pattern.

As mentioned above, we are planning to reduce the number of circulation types. Showing the pollutant anomalies for each species and AOD side by side with each circulation type would increase the number of plots. We would prefer to explain briefly towards the end of the section how the different species relate with one another.

**Specific Comments:**

Line 141: Please define TqJ

- Clarified in the revised text. TqJ signifies the joint temperature and humidity retrievals. These are recommended to be used for the process and climate studies.

Line 167: Why is the weighting needed? To ensure each month of spring receives equal weight?
- Yes. Since the number of events in each month are different and also depend on the circulation type, the weighting ensures that the climatology also reflects this event distribution.

Line 171: It is stated here that only statistically significant anomalies are shown, but some figures (like Fig. 3) appear to show anomalies everywhere. How is significance or non-significance indicated?
- We chose not to mask the anomalies of meteorological variables (in Figs. 2, 3 and 4) based on the statistical significance. This is to facilitate the interpretation of the selected circulation types and the better understanding of the transport patterns. However, we decided to show only the statistically significant anomalies when presenting the anomalies of the chemical pollutants ($NO_2$, CO and $O_3$) and AOD.

Line 206: What does "those circulation types" refer to?
- The term refers to the 20 circulation types considered in the manuscript. In the revised manuscript, the number of circulation types will be reduced to 8, see comment above (General comment #3).

Lines 268-295: I find it difficult to relate this discussion to the large number of alternating positive and negative anomalies that appear in Fig. 7. Perhaps the analysis would be more convincing if multiple circulation types were grouped together to improve sample size and data coverage.
- See answer to one of the General Comments above.

Fig. 1: A discrete colorbar might be easier to interpret.
- A discrete colorbar is used and the figure is revised.

Fig. 2: Streamlines might be a nice addition to help visualize the direction of transport
- Streamlines are added in Fig. 2.

References:

Devasthale, Abhay & Sedlar, Joseph & Tjernström, Michael & Kokhanovsky, Alexander. (2020). A Climatological Overview of Arctic Clouds. 10.1007/978-3-030-33566-3_5.

Huijnen, V., Miyazaki, K., Flemming, J., Inness, A., Sekiya, T., and Schultz, M. G.: An intercomparison of tropospheric ozone reanalysis products from CAMS, CAMS interim, TCR-1, and TCR-2, Geosci. Model Dev., 13, 1513–1544, https://doi.org/10.5194/gmd-13-1513-2020, 2020.

Inness, A., Ades, M., Agustí-Panareda, A., Barré, J., Benedictow, A., Blechschmidt, A.-M., Dominguez, J. J., Engelen, R., Eskes, H., Flemming, J., Huijnen, V., Jones, L., Kipling, Z., Massart, S., Parrington, M., Peuch, V.-H., Razinger, M., Remy, S.,

Schulz, M., and Suttie, M.: The CAMS reanalysis of atmospheric composition, Atmos. Chem. Phys., 19, 3515–3556, https://doi.org/10.5194/acp-19-3515-2019, 2019.

Johansson, E., Devasthale, A., Tjernström, M., Ekman, A. M. L., & L'Ecuyer, T. (2017). Response of the lower troposphere to moisture intrusions into the Arctic. Geophysical Research Letters, 44, 2527– 2536. https://doi.org/10.1002/2017GL072687

Schultz, M. G., Schröder, S., Lyapina, O., et al..: Tropospheric Ozone Assessment Report, links to Global surface ozone datasets, PANGAEA, https://doi.org/10.1594/PANGAEA.876108 , 2017b.

Schultz, M. G., Schröder, S., Lyapina, O., et al.: Tropospheric Ozone Assessment Report: Database and Metrics Data of Global Surface Ozone Observations, Elem. Sci. Anth., 5, 58, https://doi.org/10.1525/elementa.244, 2017a.

Schultz, M. G., Schröder, S., Lyapina, O., et al..: Tropospheric Ozone Assessment Report, links to Global surface ozone datasets, PANGAEA, https://doi.org/10.1594/PANGAEA.876108 , 2017b.

Thomas, M. A., Devasthale, A., Tjernström, M., & Ekman, A. M. L. (2019). The relation between aerosol vertical distribution and temperature inversions in the Arctic in winter and spring. Geophysical Research Letters, 46, 2836– 2845. https://doi.org/10.1029/2018GL081624

Citation: https://doi.org/10.5194/acp-2021-458-RC1

---

## Author Comment (AC2)

**Response to Referee #2**

This is a study about pollutant distributions in the Arctic, as they relate to the atmospheric circulation patterns in the springtime. SLCPs; O3, NO2, CO, and aerosols (via AOD) were examined, and correlations were found between an increase/decrease in pollutants and types of circulation. Satellite (OMI, AIRS, CALIPSO) and reanalysis (CAMS, ERA5) datasets were used, and the 20 circulation patterns were determined using a Self-Organizing Map method for the 2007-2018 time period. O3 concentrations were found to have the opposite behavior as that of NO2 for the circulation types, and NO2 was found to be the most sensitive to circulation type than the other pollutants.

We thank the referee for the constructive comments that lead to the improvement of the manuscript. Please find below point by point reply to them.

**General comments:**

1. Lines 151-162: Can you please add an explanation or justification on why mean sea-level pressure is the only variable needed to characterize a distinct circulation pattern?

MSLP is a robust indicator of an atmospheric state in the Arctic, and captures and represents the circulation and flow patterns that affect the lower troposphere (Neal et al., 2016 and the references therein). This is important for studying the pollution transport processes that occur mostly in the lower troposphere and their subsequent impacts. We have also used geopotential height anomalies to see the coupling of the upper troposphere with the lower troposphere and we can clearly see that for the most part they agree with one another indicating a coupling between the upper and lower troposphere during the various circulation regimes investigated here. A discussion about this is added in the revised manuscript.

2. From Fig 1 – if I interpret it correctly, CT#20 is the most frequently occurring CT in March. And CT#1 & 4 are the most commonly occurring CTs in May. CT#6 and 7 are not very frequent in any of the 3 months. It would be helpful if the authors spent some time discussing which of the 20 CTs are the most common conditions and which are more rare and to further discuss that frequency in terms of the SLCP concentrations. If I've misunderstood and all 20 CTs have a similar frequency of occurrence, than the authors should explain that too.

We thank the referee for the suggestion and we agree completely with it. We have added a paragraph discussing the frequency of different circulation types in the revised manuscript. The CTs do not have similar frequency and the number of events in CT differs as well in each month. Therefore the weighting factor was used to compute the climatology to compute the unbiased anomalies.

In the revised manuscript, we have kept only 8 circulation types to avoid redundancy (please see our response to the similar issue raised by Referee #1). We hope this will improve the readability of the manuscript.

**Minor comments:**

Line 18: spell out acronym "MSLP".
- Mean Sea Level Pressure (MSLP) is spelled out.

Line 21: spell out acronym "AOD".

- Aerosol Optical Depth (AOD) is spelled out.

Line 94: "descend" should be "descent"?
- Corrected.

Figure 1: The captions says (a) and (b), but the figure panels aren't labelled with (a) and (b) but they should be. Otherwise, the caption should be changed to (top) and (bottom).
- Corrected.

The x-axis of both panels should be labelled ("number of days"?). The colour bar or the the lower panel is labelled "circulation type number", but I think it should instead be labelled "weighing factor", no?
- The X-axis in both subplots show the circulation type number, while the subplots themselves show the number of days each circulation type has occurred (top) and the corresponding weighting factor (bottom).

Line 262: add "and" between 'humidity, rainfall'.
- Corrected.

Figure 6: what's the unit on the O3 anomalies? Is it unitless VMR? Perhaps multiply by 10^9 and provide units of ppbv. Or else, add to the caption as you've done in Fig 8 ("The … volume mixing ratio anomalies…")
They are VMR. They are expressed in ppbv in the revised manuscript.

References:
Neal, R., Fereday, D., Crocker, R. and Comer, R.E. (2016), A flexible approach to defining weather patterns and their application in weather forecasting over Europe. Met. Apps, 23: 389-400. https://doi.org/10.1002/met.1563

Citation: https://doi.org/10.5194/acp-2021-458-RC2

---

## Author Response (AR2)

**RC1- Second review: 'Comment on acp-2021-458', Anonymous Referee #1**

The revised text and figures strengthen the paper, which offers interesting insights into the effect of atmospheric circulation patterns on pollutants in the Arctic. I offer a couple of suggestions/questions related to the new text:

1. It would be useful to indicate somewhere, perhaps on the corresponding figures in the appendix, which of the 20 patterns in the appendix correspond to the 8 shown in the main text.
The following sentence is added to the manuscript " The 8 CTs (CT1 - CT8) selected in this study (shown in Fig. 2) correspond respectively to CT1, CT4, CT9, CT12, CT14, CT18, CT19 and CT20 in Fig. A1.1 in the Appendix."

2. On page 8, it is stated: "An inverse correspondence between O3 and NO2 away from the source regions is not expected due to the different life times, aging and transport processes." Can you speculate on what causes this unexpected result? Do photolysis or stratosphere-troposphere exchange play a role?
The following paragraph in the manuscript speculates this result.
"The springtime photochemistry in the Arctic is very complex, as duly noted in the rich literature that documents the research and observations on this subject matter (Lu et al., 2019 and the references therein). The interactions between $NO_2$ and $O_3$ are also highly non-linear in reality and hence a one-one correlation can not be established. In the troposphere, NO is converted to $NO_2$ in the presence of $O_3$ which is a potential sink for $O_3$. However, during sun-lit conditions, $NO_2$ is converted back to NO via photolysis which results in $O_3$ production. Apart from these chemical reactions, local meteorological conditions such as temperature, relative humidity and rainfall play an important role in the production and dispersion of these pollutants. Stratospheric intrusions are another source of $O_3$ variability in the troposphere that may play a role under different circulation types (Yates et al., 2013; Langford et al., 2015; Lin et al., 2015). The persistent anticyclonic conditions could not only lead to the accumulation of the tropospheric $O_3$, but also favour the large-scale descent or intrusions into the lower troposphere, leading to positive $O_3$ anomalies."